# Rapid and Simple Detection of Ochratoxin A using Fluorescence Resonance Energy Transfer on Lateral Flow Immunoassay (FRET-LFI)

**DOI:** 10.3390/toxins11050292

**Published:** 2019-05-23

**Authors:** Hyun-Kyung Oh, Hyou-Arm Joung, Minhyuk Jung, Hohjai Lee, Min-Gon Kim

**Affiliations:** 1Department of Chemistry, Gwangju Institute of Science and Technology, 123 Cheomdangawgi-ro, Buk-gu, Gwangju 61005, Korea; hkoh@gist.ac.kr (H.-K.O.); ammy2002@ucla.edu (H.-A.J.); mhjung817@gist.ac.kr (M.J.); hohjai@gist.ac.kr (H.L.); 2Electrical & Computer Engineering Department, University of California, Los Angeles, CA 90095, USA; 3INGIbio Co. Ltd., R&D Center, 206, APRI, 123 Cheomdangawgi-ro, Buk-gu, Gwangju 61005, Korea

**Keywords:** ochratoxin A (OTA), mycotoxin, fluorescence resonance energy transfer (FRET), lateral flow immunoassay (LFI), label-free, matrix effect

## Abstract

The detection of mycotoxins is crucial because of their toxicity in plants, animals, and humans. It is very important to determine whether food products are contaminated with mycotoxins such as ochratoxin A (OTA), as mycotoxins can survive heat treatments and hydrolysis. In this study, we designed a fluorescence resonance energy transfer (FRET)-based system that exploits antibody-antigen binding to detect mycotoxins more rapidly and easily than other currently available methods. In addition, we were able to effectively counteract the matrix effect in the sample by using a nitrocellulose membrane that enabled fluorescence measurement in coffee samples. The developed FRET on lateral flow immunoassay (FRET-LFI) system was used to detect OTA at a limit of detection (LOD) of 0.64 ng∙mL^−1^, and the test can be completed in only 30 min. Moreover, OTA in coffee samples was successfully detected at a LOD of 0.88 ng∙mL^−1^_,_ overcoming the matrix effect, owing to the chromatographic properties of the capillary force of the membrane. We believe that the developed system can be used as a powerful tool for the sensitive diagnosis of harmful substances such as mycotoxins and pesticides for environmental and food quality control monitoring.

## 1. Introduction

Mycotoxins are secondary metabolites produced by fungi that can cause diseases or abnormalities in humans and animals [1,2,3]. The toxic effect of these metabolites in mammals is known as mycotoxicosis [4,5]. Mycotoxins usually contaminate grains or nuts, which are useful hosts for the breeding of molds [6,7]; they are highly influenced by environmental factors such as temperature, moisture, and rainfall [8,9]. Mycotoxins including ochratoxin A (OTA), ochratoxin B and C (OTB, OTC), aflatoxin, patulin, zearalenone (ZEA), and others are produced by various species within the genera *Fusarium*, *Aspergillus*, and *Penicillium* [10]. Among them, OTA has recently gained a lot of attention because of its biological toxicity—it can cause effects such as teratogenicity, carcinogenicity, immune toxicity, and even fertility inhibition [11,12]. OTA is also related to a human disease known as Balkan endemic nephropathy, which is mainly found in Southeastern Europe [13]. The detection of OTA in food products is important because the ingestion of OTA-contaminated food constitutes a risk to human and animal health. These concerns have promoted the development of simple, rapid, and on-site analytical methods.

Many different analytical methods are now available to detect OTA, including thin layer chromatography (TLC) [14], gas chromatography with mass spectroscopy (GC-MS) [15], high performance liquid chromatography (HPLC) [16], liquid chromatography with mass spectroscopy (LC-MS) [17], real time polymerase chain reaction (RT-PCR) [18], and enzyme-linked immunosorbent assays (ELISA) [19]. These are the most common methods are used to detect OTA; however, each of these methods has a limitation, such as low sensitivity, the need for sophisticated instruments or expertise, or the time required to perform the test. 

Recently, a membrane-based test strip, also referred to as the lateral flow immunoassay (LFI), has become popular for diagnostic tests [20,21,22]. The reason for its popularity is that the device provides a rapid, simple, low-cost, and easy to operate detection method. This device does not require any further reagents to obtain results after applying the sample solution, and users can read the results quickly and easily without the need for specialized instruments or expertise. Several studies have attempted the analysis of OTA using a membrane-based immunosensor [23,24,25,26,27,28]. However, most current methods for OTA analysis require additional labeling steps, which are complicated and time-consuming.

In 2011, a homogeneous immunoassay for the detection of OTA was reported by Li et al. [29]. They developed a label-free, direct, and noncompetitive homogeneous immunoassay by detecting the fluorescence resonance energy transfer (FRET) between OTA and an anti-OTA antibody. This method shows a remarkable ability to detect mycotoxins, but is still limited in several ways; importantly, darkly colored solutions, including coffee and red wine, cannot be applied to this system as the fluorescence signal will not be detectable in a dark solution because of the matrix effect. To overcome this limitation, the development of an advanced novel assay is required. 

In this study, we applied a label-free, direct, and non-competitive FRET system to the lateral flow immunoassay (LFI) to develop a novel platform for the easy and rapid detection of OTA. We aimed to achieve the following in our developed FRET-LFI system: (1) a rapid and simple label-free immunoassay system; (2) a signal-on quantitative method for small molecule detection; and (3) a system in which the matrix effect is negated in order to apply this system directly to real samples, especially darkly colored solutions. Thus, the developed FRET-LFI should be a powerful tool for the rapid and sensitive quantification of mycotoxins in the field.

## 2. Results and Discussion

### 2.1. Optimization of the Developed FRET-LFI for OTA Analysis

The developed FRET-LFI in this study was designed based on the FRET between OTA and the anti-OTA antibody. Anti-OTA antibodies were immobilized on a nitrocellulose membrane, and the sample solution containing the target antigen was allowed to flow from the bottom of the sensor, as shown in Figure 1a. The sample solution flowed over the anti-OTA antibody immobilized membrane due to the capillary force of the membrane, and the target material in the sample solution reacted with the antibody immobilized on the membrane. After the immune reaction, the fluorescence spectrum could be identified by irradiating the membrane at 280 nm to determine antigen-antibody binding. As shown in Figure 1b, when the antibody was alone on the membrane, an intrinsic fluorescence emission band at 330 nm could be detected upon excitation at 280 nm. The antigen alone resulted in an intrinsic fluorescence emission band at 440 nm upon excitation at 330 nm. However, the antibody and the antigen together would produce emission bands at both 330 nm and 440 nm upon excitation at 280 nm because of the FRET between the antibody and the antigen. Applying this system to a membrane allows matrix effects to be minimized, and fluorescence signals could be identified even in dark substances, such as coffee, enabling the rapid and easy analysis of OTA.

To confirm the FRET phenomenon between the antibody and the antigen on the membrane, different concentrations of OTA were applied to membranes containing the immobilized antibody, and their fluorescence emission spectra were scanned. The fluorescence images are shown in Figure 2a with a bandpass filter (450/40) upon excitation at 254 nm from an ultraviolet lamp, and the fluorescence spectra were different depending on the concentration of OTA (Figure 2b–d). As the concentration of OTA increased—the intensity of the emission at 330 nm, the excitation wavelength of the antibody—decreased, while that at 440 nm, the excitation wavelength of the antigen increased.

### 2.2. Sensitivity and Selectivity of FRET-LFI

The developed FRET-LFI optimized in this study was evaluated by testing its sensitivity using standard OTA solutions at concentrations of 0, 1, 10, 100, and 1000 ng∙mL^−1^ in methanol/4-morpholineethanesulfonic acid hydrate (MES)-tween 20 (10:90, *v*/*v*), and the specificity of the assay was studied by adding other mycotoxins at a concentration of 10 ng∙mL^−1^. The sensitivity of the dipstick assay is shown in Figure 3. Increasing the concentration of OTA decreased the intensity of the emission at 330 nm and increased that at 440 nm. We fitted the data to a sigmoidal curve, and obtained well-correlated results (*R*^2^ = 0.96). The limit of detection, defined as sum of blank signal and standard deviation 2× from the OTA in the FRET-LFI, was estimated using the curve formula. The specificity of the dipstick assay was further examined by exposing the membrane to the mycotoxins OTB and ZEA (Figure 4). No significant increase in the fluorescence intensity was observed when ZEA was introduced to the solution. In contrast, the system showed some reactivity to OTB because its structure is similar to that of OTA; however, it was confirmed that the specificity for OTA was greater. This issue will be addressed and improved in a future study. These results suggest that the developed FRET-LFI can be used for the sensitive and selective detection of OTA. 

### 2.3. Analysis of OTA-Spiked Coffee Samples

The matrix effect refers to the effects on an analytical tool caused by all components of the sample other than the target material; this phenomenon has long been a problem in analytical technology. The membrane applied for the sensing platform developed here has chromatographic properties that can reduce the matrix effect and enable the target material to be measured by fluorescence in dark solvents such as coffee. This phenomenon was evaluated by applying an OTA-spiked coffee solution to the developed assay system. OTA-spiked coffee solutions containing concentrations of 0, 1, 10, 100, and 1000 ng∙mL^−1^ OTA were applied to the anti-OTA antibody-immobilized membrane, and the fluorescence intensity was measured at 330 nm and 440 nm. As shown in Figure 5, it was confirmed that OTA was clearly detected at all concentrations, even though coffee is a dark solution that normally cannot be used for fluorescence detection methods. We fitted the data to a sigmoidal curve, and obtained well-correlated results (*R*^2^ = 0.99). The limit of detection, defined as sum of blank signal and standard deviation 2× from the OTA in the FRET-LFI, was estimated using the curve formula. The results indicated that the matrix effect was successfully negated in our FRET-LFI sensor, and that this system can be applied to detect other small molecules found in dark solutions such as coffee, wine, beer, fruit juice, among others.

## 3. Conclusions

The permitted levels of OTA, a common mycotoxin, are severely restricted in food products because the ingestion of food contaminated with this toxin is harmful [30]. Therefore, various methods for the rapid and simple detection of OTA in foods and beverages are being developed. In this study, we developed a rapid and simple FRET-LFI sensor for the sensitive and accurate detection of OTA. Although this sensor requires a multi-step process to operate, including a washing step, all steps can be completed within 30 min, and the results can be measured immediately. In particular, owing to the nature of the developed sensor, the matrix effect was eliminated, allowing fluorescence results to be successfully confirmed even in dark solvents such as coffee. The detection limits of the developed dipstick assay were 0.64 ng∙mL^−1^ OTA in buffer and 0.88 ng∙mL^−1^ OTA in a coffee sample; furthermore, the system shows high selectivity to OTA, but not other mycotoxins. The FRET-LFI sensor developed in this study has the potential to be applied to other analysis methods required for the rapid and simple analysis of small molecules such as mycotoxins.

## 4. Materials and Methods 

### 4.1. Chemicals and Reagents

OTA, OTB, ZEA, MES hydrate, tween 20, and methanol were purchased from Sigma Aldrich (St. Louis, MO, USA), and 96-microwell plates (U bottom) were purchased from SLP Life Sciences (Pocheon, Korea). The anti-OTA antibody was produced as previously described [31]. Nitrocellulose membranes and absorbent pads were purchased from Millipore Co. (Bedford, MA, USA) and Ahlstrome-Munksjö (Helsinki, Finland), respectively. Coffee was purchased from a local coffee shop (Gwangju, Korea). 

### 4.2. Preparation of the Fluorescence Immunoassay System for OTA Detection

For the OTA immunoassay, 0.8 µL anti-OTA solution (0.44 mg∙mL^−1^ in 2 mM Borate buffer, pH 8.5) was immobilized on a nitrocellulose membrane, and the antibody-loaded membrane was dried at 37 °C for 15 min. An absorbent pad was then attached on top of the membrane with a 2-mm overlap. The prepared nitrocellulose membrane was placed on top of a 96-well microplate, and the immobilized portion of the antibody was centered in the well. Next, 10 µL standard OTA solution in methanol was added to 90 µL 25 mM MES buffer (pH 6.0) containing 0.3% tween 20. The mixture was added to the membrane and incubated at room temperature for 10 min. To remove any unbound OTA antigens, 70 µL 25 mM MES buffer (pH 6.0) containing 0.3% tween 20 was then applied to the membrane and incubated at room temperature for a further 10 min. Finally, the membrane was dried completely at 37 °C for 10 min.

### 4.3. Determination of the Specificity of the Fluorescence Immunoassay System

To determine the specificity of the LFI system, a dipstick assay was conducted by analyzing OTA and other mycotoxins, OTB and ZEA, at a standard concentration of 10 ng∙mL^−1^. The OTA and other mycotoxins in methanol were prepared with 25 mM MES buffer (pH 6.0) containing 0.3% tween 20. The solutions were applied to the antibody-immobilized membrane and incubated at room temperature for 10 min. To remove any unbound OTA or other mycotoxins, 70 µL 25 mM MES buffer (pH 6.0) containing 0.3% tween 20 was then applied to membrane and incubated at room temperature for a further 10 min. Finally, the membrane was dried completely at 37 °C for 10 min.

### 4.4. Preparation of Spiked Coffee Samples

OTA-positive coffee samples to validate the designed LFI system were prepared by spiking OTA-free coffee samples with OTA at known concentrations of 0, 1, 10, 100, and 1000 ng∙mL^−1^. Next, 25 mM MES (pH 6.0) with 0.3% tween 20 was added to the OTA-positive coffee samples. The sample solutions were applied directly to the LFI system and assayed as described above. OTA-free coffee samples were defined in the sample as "with an undetectable level of OTA according to the FRET-LFI" since there was no difference value of results between the buffer-only sample and the OTA-free coffee sample. Since the coffee sample was prepared as a liquid type, it was difficult to determine the OTA content (ng∙mL^−1^) of roasted coffee beans. Following the Risk Assessment of Mycotoxins released in 2016 by the Ministry of Food and Drug Safety in Korea, however, OTA is also detectable with the limit of detection at 0.5 ng∙mL^−1^ in coffee beverages.

### 4.5. Fluorescence Imaging on a Nitrocellulose Membrane

For fluorescence imaging, a nitrocellulose membrane was placed on an upright epifluorescence microscope (OPTIPHOT-100, Nikon, Tokyo, Japan), and directly illuminated by a 254-nm ultraviolet (UV) lamp (LF-206.LS, UVITEC, Milton, England) from the side. The fluorescence emitted from the membrane was collected by a 2× objective (PlanApo N 2×, NA 0.08, Olympus, Tokyo, Japan), and imaged on an electron multiplying charged coupled device (EMCCD) camera (iXon 897, Andor, Abingdon-on-Thames, England). An emission filter centered at 450 nm (FB450-40, Thorlabs, Newton, NJ, USA) was placed in front of the camera to separate fluorescence from scattered illumination light. Two images were taken for each samples, IM_R_(*x*,*y*) and IM_L_(*x*,*y*), which correspond to the fluorescence images illuminated by UV lamp from the left and right side, respectively. An integrated image, IM(*x*,*y*), was then obtained by merging both images—IM_R_(*x*,*y*) and IM_L_(*x*,*y*). The processing of the image was conducted by MATLAB software (R2013b, MathWorks, Natick, MA, USA).

### 4.6. Instrumentation

A drying oven (KO0199; LK Lab Co., Namyangju, Korea) was used to dry the membranes after they were loaded with antibodies, and a cutting device (TBC-50Ts; Taewoo Co., Namyangju, Korea) was used to cut the nitrocellulose membranes. The fluorescence intensities of samples on the nitrocellulose membranes were determined at a range of 300–500 nm with excitation at 280 nm and 2-nm intervals using a well plate reader (Infinite M200pro; TECAN Group, Ltd., Männedorf, Switzerland). An ultra-violet lamp (LF-206.LS; UVITEC, England, UK), a microscope (OPTIPHOT-100; Nikon, Tokyo, Japan), a charged coupled device (iXon 897; Andor, Ulster, UK), a bandpass filter (FB450-40; Thorlabs, Newton, NJ, USA), an objective (PlanApo N 2X; Olympus, Tokyo, Japan), and a software program (Andor-SOLIS 64-bit; OXFORD INSTRUMENTS, Abingdon-on-Thames, UK) were used to capture the fluorescence image in a nitrocellulose membrane.

## Figures and Tables

**Figure 1 toxins-11-00292-f001:**
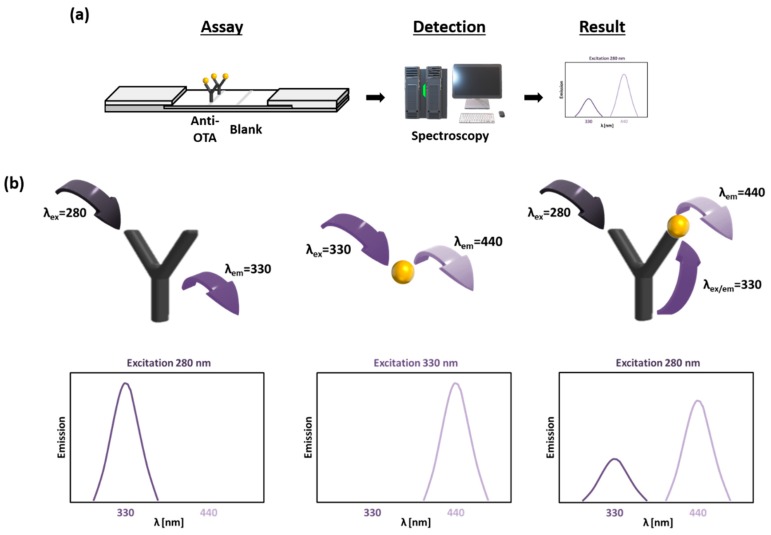
Schematic illustration of the developed fluorescence resonance energy transfer on lateral flow immunoassay (FRET-LFI) sensor. (**a**) The procedures and (**b**) a scheme of the FRET-LFI sensor for an anti-OTA antibody only (left), an antigen only (middle), and both (right) are presented. OTA: ochratoxin A.

**Figure 2 toxins-11-00292-f002:**
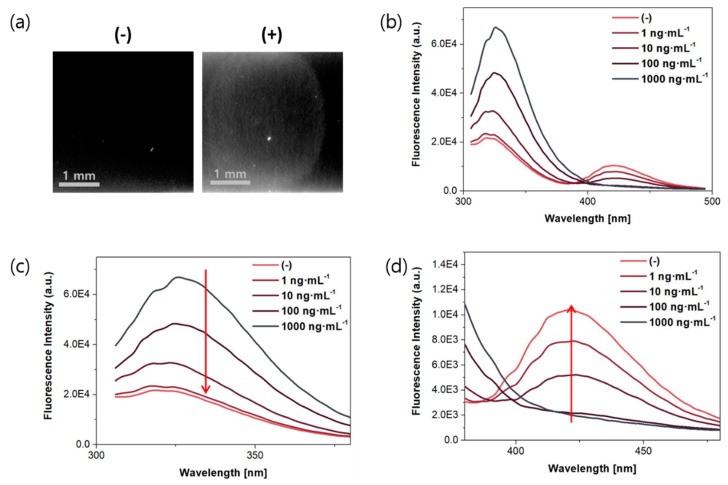
Fluorescence images and emission spectra of anti-OTA antibody (0.44 mg∙mL^−1^) upon exposure to different concentrations of OTA in 25 mM 4-morpholineethanesulfonic acid hydrate (MES) buffer (pH 6.0) with 0.3% tween 20 (*λ*_ex_ = 280 nm). (**a**) Fluorescence images in a nitrocellulose membrane without OTA (left) and with OTA (right) with a bandpass filter (450/40) (*λ*_ex_ = 254 nm), and fluorescence spectra in different wavelengths (**b**) from 300 to 500 nm; (**c**) from 300 to 380 nm; and (**d**) from 370 to 480 nm.

**Figure 3 toxins-11-00292-f003:**
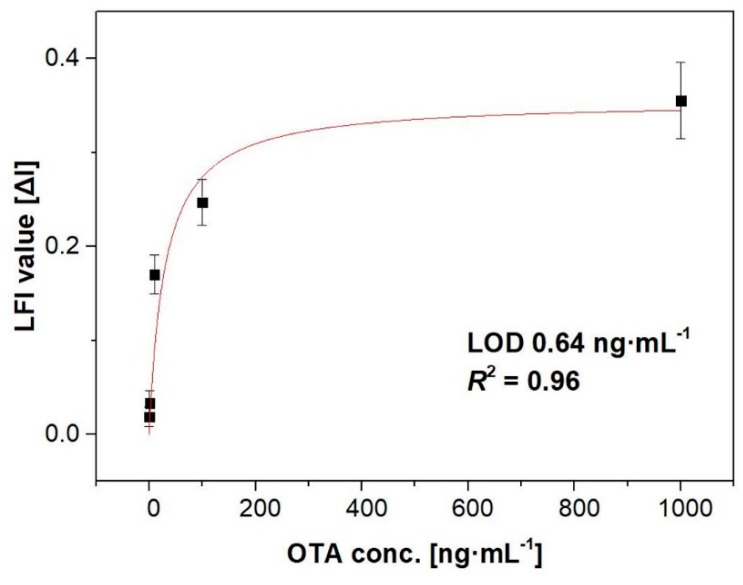
Calibration curve showing the 440/330 nm ratios of fluorescence intensity at different standard OTA concentrations in buffer. The error bars indicate the standard deviation from three independent experiments. LOD: limit of detection.

**Figure 4 toxins-11-00292-f004:**
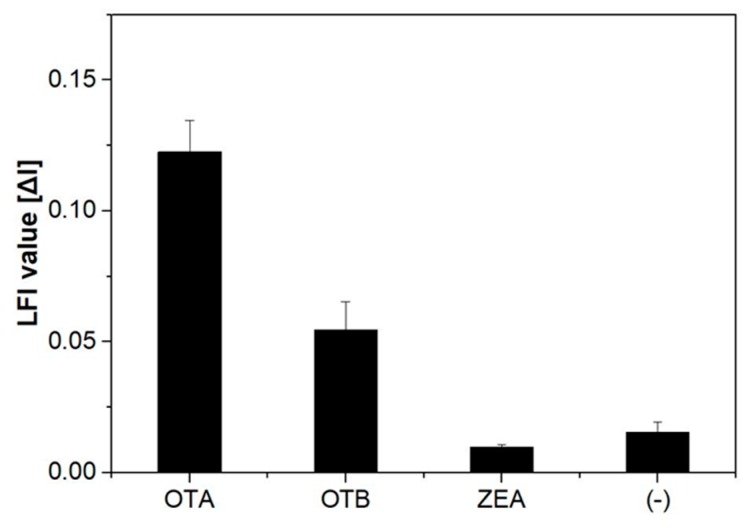
Specificity of the system to the mycotoxins OTA, ochratoxin B (OTB), and zearalenone (ZEA) at a standard concentration (10 ng∙mL^−1^). The error bars indicate the standard deviation from three independent experiments.

**Figure 5 toxins-11-00292-f005:**
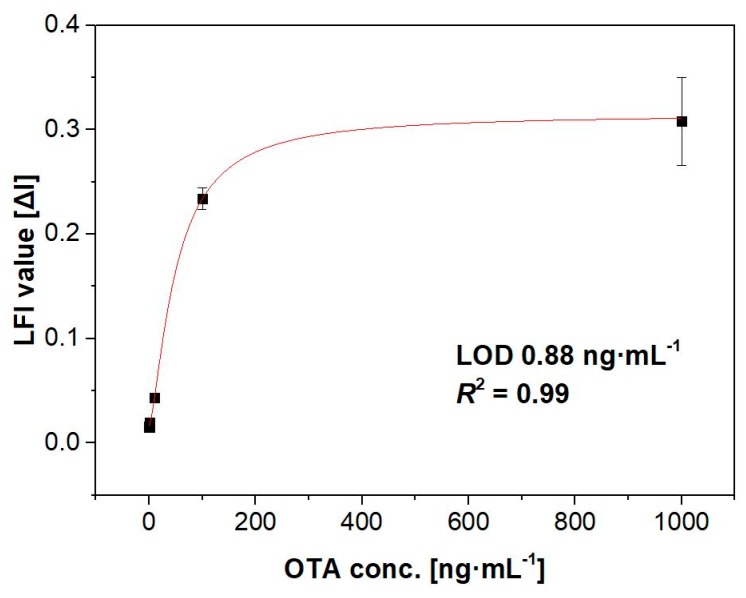
Calibration curve showing the 440/330 nm ratios of the fluorescence intensity at different concentrations of OTA in coffee. The error bars indicate the standard deviation from three independent experiments. LOD: limit of detection.

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
