# Peer review of "Rapid and Simple Detection of Ochratoxin A using Fluorescence Resonance Energy Transfer on Lateral Flow Immunoassay (FRET-LFI)"

_toxins, 2019, doi:10.3390/toxins11050292_

Round 1

Reviewer 1 Report

It is necessary to describe better, how were the coffee samples treated. In the part 4 (Materials and Methods) is written, that coffee was purchased from a local coffee shop.

1) Which type of coffee - roasted beans?  

2) How it was proved that the coffee samples were OTA-free? (stated in line 192). Add explanation.

3) How was the coffee (solution) prepared? The relationship between the content of OTA in roasted coffee beans  (ng/mg) and in the prepared solution (as you stated – in ng/ml) is not clear. It is important, as the limits for OTA are set in roasted coffee, not in the beverage.

Line 28-29: The sentence doesn´t make proper sense: „Mycotoxins usually contaminate grains or nuts, which are useful hosts for the breeding of molds; …“

Line 78:  link to Figure 1c – there is no Figure 1c.

Line 175:  abbreviation NC is used , but not explained. Probably it means nitrocellulose (membrane ). But this word is used previously (line 72, 167,  173) and  thereafter (line 196, 209, 210, 216,.. etc) without abbreviation. It should be set right

Line 205: Please specify the MATLAB software, and the city a state of the company.

Author Response

Point 1: Which type of coffee - roasted beans?

Response 1: The type of coffee was liquid type from roasted beans.

Point 2: How it was proved that the coffee samples were OTA-free? (stated in line 192). Add explanation.

Response 2: OTA-free coffee was tested against the buffer before OTA spiking. When we compare the result with the buffer that does not contain OTA, no difference in value was shown. Therefore, it was considered that the coffee sample had no OTA.

Point 3: How was the coffee (solution) prepared? The relationship between the content of OTA in roasted coffee beans (ng/mg) and in the prepared solution (as you stated – in ng/ml) is not clear. It is important, as the limits for OTA are set in roasted coffee, not in the beverage.

Response 3: It is difficult to determine the OTA content (ng / mg) of roasted coffee beans, since coffee samples were prepared in liquid form, and spiked OTA in ng / mL units in the solution. However, in the Risk Assessment of Mycotoxins released in 2016 by Ministry of Food and Drug Safety in Korea, we can confirm that OTA is detected at 0.5 ng / mL in coffee liquid solution. The actual OTA content (ng / mL) in roasted coffee beans will be discussed in the further study.

Point 4: Line 28-29: The sentence doesn´t make proper sense: “Mycotoxins usually contaminate grains or nuts, which are useful hosts for the breeding of molds; …
”

Response 4: The phrase what we would explain means that the molds grows easily in grains or nuts.

Point 5: link to Figure 1c – there is no Figure 1c.

Response 5: I have changed the word 1c to 1b.

Point 6: abbreviation NC is used, but not explained. Probably it means nitrocellulose (membrane). But this word is used previously (line 72, 16, 173) and thereafter (line 196, 209, 210, 216,.. etc) without abbreviation. It should be set right

Response 6: The words, NC and nitrocellulose, have been homogenized to the word “nitrocellulose.”

Point 7: Please specify the MATLAB software, and the city a state of the company.

Response 7: I have specified the MATLAB software (R2013b), and added the city of the company.

Reviewer 2 Report

In this manuscript, Authors introduced the OTA detection method by using FRET-LFI. They suggested the LOD of 0.64 nm/mL and demonstrated the detection of OTA in a coffee sample.  The manuscript is generally well-written and the conclusions are supported by the results. Therefore, I recommend this manuscript to accept in Toxins after minor revision.

1) It will be nice to discuss the comparison of this method with previously reported OTA sensing methods.

Author Response

Point 1: It will be nice to discuss the comparison of this method with previously reported OTA sensing methods.

Response 1: In line 47, common methods for analysing OTA are listed, and previous reports for detecting OTA using a membrane sensor are listed on line 54. This study was conducted to overcome the limitation from those analytical methods.
